# Bilateral Acute Macular Neuroretinopathy in a Young Patient: Imaging and Visual Field during Two-Year-Follow-Up

**DOI:** 10.3390/diagnostics10050259

**Published:** 2020-04-28

**Authors:** Alessandro Porta, Sarah Tripodi, Mario Damiano Toro, Robert Rejdak, Konrad Rejdak, Emma Clara Zanzottera, Fabio Ferentini

**Affiliations:** 1Department of Ophthalmology, Hospital C. Cantù, 20081 Abbiategrasso, Italy; 2Department of Ophthalmology, IRCCS Sacro Cuore Don Calabria Hospital, Via Don Sempreboni 5, 37024 Negrar, Italy; 3Department of General Ophthalmology, Medical University of Lublin, 20079 Lublin, Poland; 4Faculty of Medical Science, Collegium Medicum Cardinal Stefan Wyszyński University, 01815 Warsaw, Poland; 5Department of Neurology, Medical University of Lublin, 20079 Lublin, Poland; 6Department of Ophthalmology, San Gerardo Hospital, 20900 Monza, Italy

**Keywords:** acute macular neuroretinopathy, optical coherence tomography (OCT), imaging, external retinal layers

## Abstract

Acute macular neuroretinopathy (AMN) is a rare disorder. We report a case of bilateral AMN in a young female patient, without any risk factors. She referred a positive scotoma in both eyes after flu-like symptoms. Fundus examination revealed parafoveal dark-reddish oval lesions in both eyes. Therefore, we performed visual field, optical coherence tomography (OCT), fluorescein angiography (FA) and indocyanine green angiography (ICG) at baseline and several times during the two years of follow-up. The infrared (IR) imaging showed one rounded hyporeflective lesion in the left eye and two similar lesions in the right eye. The OCT demonstrated the characteristic alterations in the outer retina. The visual field also demonstrated scotomas corresponding with these lesions. The OCT and IR features disappeared at the end of the follow-up except for the left eye, which continued to have hyperreflective spots in the outer plexiform layer. The patient complained about a residual scotoma only in the left eye after two years. Our case shows a difference in disease progression in the two eyes of the same patient, suggesting that several mechanisms can be implicated in the pathology of AMN.

## 1. Introduction

Acute macular neuroretinopathy (AMN) is a rare disorder with an unclear etiology. It was described for the first time by Bos and Deutman in 1975 [1]. Usually it occurs in young Caucasian female patients, during their reproductive years [2,3]. AMN was initially linked with contraceptive pill use [1], but thereafter it has been reported in several conditions, including shock, trauma, eclampsia, epinephrine, sympathomimetic or cocaine use, hypovolemia and heavy coffee intake, dengue fever, and systemic lupus erythematosus [4,5,6,7,8,9,10,11,12].

Typical symptoms are photopsias and central or paracentral scotomas, usually bilateral, that occur suddenly in these patients [2]. Occasionally these symptoms can also be preceded by a viral prodrome [13]. The diagnosis of AMN may be difficult because of the subtle or absent findings on fundus examination and fluorescein angiography (FA) [14].

Recently, authors have identified ischemia involving the deep retinal capillary plexus as a possible pathogenic mechanism using spectral-domain optical coherence tomography (SD-OCT) [15].

It has been shown in previous studies that retinal lesions in AMN represent circumscribed areas of hypoperfusion that lead to atrophy of the outer nuclear layer in these areas, in contrast to paracentral acute middle maculopathy (PAMM) where the inner layers are involved [16]. In particular, the AMN lesions may develop at the junction of the outer plexiform layer (OPL) and outer nuclear layer (ONL) with associated outer macular disruption [17,18]. To support these findings, we reported the imaging and visual fields of a young patient with bilateral AMN who was followed for a period of over two years.

## 2. Case Presentation

A 21-year-old woman was referred to our department in December 2016 with one day history of fixed grey-dark spots in the vision field of both eyes, with most of those spots in her left eye. She described to the shape of these scotomas as a “tear drop”, and she was able to draw them precisely on an Amsler chart. This episode was preceded by a single day of a flu-like illness, with an elevated temperature (39 °C). Her medication history was notable solely for paracetamol. Her medical history was solely remarkable for a history of varicella-zoster infection in infancy. Moreover, she had stopped contraceptives for the previous three months. She denied any other pathologies, trauma, or travel history.

Upon admission, her best-corrected visual acuity (BCVA) was 20/20 in both eyes (OU). Color vision was normal. Slit lamp examination showed no abnormality. Fundus examination revealed parafoveal dark-reddish oval lesions in both eyes (two in the right eye and one in the left eye), corresponding to the abnormalities on the Amsler grid.

We performed visual field examination (Octopus 900 perimeter Haag-Streit Inc, Koenic, Switzerland), SD-OCT (Heidelberg Engineering, Heidelberg, Germany), FA (Heidelberg Engineering, Heidelberg, Germany) and ICGA (Heidelberg Engineering, Heidelberg, Germany), at baseline (one day after symptom presentation), seven days later, 25 days later, 40 days later, 10 weeks later, 22 weeks later, 12 months later, and 30 months later.

Scanning laser ophthalmoscopic (SLO) infrared imaging showed a hyporeflective, sharp oval area in the nasal area of the macular region in the left eye, and two hyporeflective oval areas nasally and inferiorly of the macula in the right eye. The lesion in the left eye was bigger than the two in the right eye.

At baseline, the OCT of the left eye (Figure 1A) showed a focal highly reflective band of the outer plexiform layer (OPL) extending into the outer nuclear layer (ONL), with a slight hyporeflectivity of the external limiting membrane (ELM), corresponding to the round lesion of the infrared (IR) image. We also noticed a disruption of the photoreceptors’ inner segment/outer segment (IS/OS) interface (Ellipsoid zone), and an associated alteration of the ELM and the retinal pigment epithelium (RPE). The inner retina layers did not show any modifications. 

At the same time, the right eye was noted to have two para-macular lesions in the IR images, but the OCT demonstrated less structural alterations compared to the left eye (Figure 2A). In particular, we documented two focal hyperreflective bands of the OPL and the ONL corresponding to the two lesions. An alteration of the IS/OS interface and RPE was reported only in the zone of the bigger lesion, the one localized nasally in the IR image.

One week later, the OCT and IR features did not change in the left eye (Figure 1B) but started to normalize in the fellow eye (Figure 2B). Only 40 days later, the OCT of the left eye started to show a reduction of the hyperreflective band in the ONL and the other external layers, but it showed the same hyperreflective band of the OPL (Figure 1C).

At the one year follow-up visit, regarding the left eye, there was minimal hyperreflectivity in the OPL with reduction of the reflectivity in the other layers approximating normalization (Figure 1D). In addition, the hyporeflective area noted in the SLO image was reduced. In contrast, a hyperreflective region in the OPL, corresponding with the smaller lesion, was solely noted in the right eye (Figure 2D). At the 30 months follow-up visit, the imaging alterations noted in both eyes were hyperreflective spots in the OPL with a corresponding hyporeflective band of the ellipsoid zone and an hyporeflective halo in the SLO image, both showing reduction compared to the initial visit (Figure 1E and Figure 2E).

At the end of follow-up, the patient still complained about a little scotoma in the left eye (less than at the beginning), but she did not refer any symptoms in the other eye. Interestingly, throughout the entire follow-up period, FA and ICGA were unremarkable for both eyes.

Visual field examination showed paracentral scotomas in both eyes (OS > OD; Figure 3A,B) corresponding to the rounded lesions noted in the fundus and SLO images. A bilateral paracentral decrease in retinal sensitivity was also seen.

A reduction in the area of the scotomas was noted after the first month. At the 30 months visit, the visual field of the right eye did not show any lesions (Figure 4A), while that of the left eye revealed a small relative paracentral scotoma (Figure 4B).

## 3. Discussion

Acute macular neuroretinopathy is a relatively rare condition that typically affects young women [19]. The etiology is still unknown [1]. To date, over 100 cases of AMN have been published and Bhavsar et al. in a review reported that 47.5% of described cases were associated with a non-specific flu-like illness or fever [2], but to date a strong link with the influenza virus has not been yet demonstrated [20]. One documented association was with the dengue fever [11,21]. Other possible associations such as cytomegalovirus, influenza A2, and Chikungunya, have shown weak evidence [22,23]. Just recently, Ashfaq et al., demonstrated the link between AMN and the influenza virus in four patients with a virologically confirmed infection. They suggested that the human leukocyte antigen (HLA) systems are not important in predisposing patients with influenza to AMN [24]. Other risk factors are use of oral contraceptive, antecedent trauma, or systemic shock [2]. Analysis of the reported risk factors in AMN seem to suggest a retinal vascular etiology.

Recently, Liu et al. [20] described a case of a middle age woman with OCT features similar to the one we found, but that patient underwent an influenza vaccination and was affected by Raynaud disease, in contrast to our young patient. On the other hand, Introini et al. suggested an association between the AMN and the intranasal cocaine use in a 24-year-old woman. That patient admitted the drug use few hours before the onset of symptoms. Cocaine may induce arterial spasm that can result in decreased ocular perfusion at the level of the deep capillary plexus (DCP) [9]. Conversely to these findings, our young patient did not report any drug use and had no risk factors, except for flu-like symptoms one day before the AMN presentation.

One recent manuscript highlighted that mutations in more than 100 different genes may predispose a person to severe outcomes after influenza [25], and this may be linked with the predisposition and pathogenesis of AMN [24]. According to Zhang et al., there are some candidate genes, whose proteins are implicated in the viral replication cycle and in host response stages [25]. More specifically, these genes could: (1) control the expression of oligosaccharide structures involved in binding of the virus to receptors on the host cell surface (DMBT1, MBL2, MUC1, SFTPA, SFTPB) [26,27]; (2) control cleavability of hemagglutinin by host proteases [28]; (3) promote the viral replication into the host cell (phase 3 genes- NS1, PABPN1, EIF2C2, SiRNA) [29,30]; (4) lead to the activation of many pro-apoptotic genes (including those encoding Fas, TP53, JUN, and CASP8) [31]; (5) up-regulate different interferon-stimulated genes (such as CXCL10, MX1, PKR, OAS1, RNASEL, and PML) [32]; and/or (6) up-regulate the expression of molecules such as PTAFR (platelet activating factor receptor) used by bacteria as receptors [33,34], resulting in a bacterial over infection. It seems that if these genes mutate, they could cause a severe increase of the inflammatory response, due to their role in the dysregulation of the host immunocompetence.

As suspected by Ashfaq et al. [24], we believe that a response of the human body to viral infections could generate an alteration of several homeostatic genic mechanisms resulting in a secondary overproduction of inflammation and an increase of apoptosis and tissue damage. Thus, in some predisposed individuals, these could lead to a damage of the retinal vascularization and consequentially to a neuroretinopathy. 

We know that influenza virus is common, instead of AMN; therefore, a genetic predisposition might exist. Future studies are needed to confirm is there is a real link between some specific genes and AMN during virus infection.

As reported in the literature, IR imaging has a crucial role in highlighting the characteristic macular lesions of AMN, and OCT has facilitated identification and classification of retinal features [14]. In our patient, fundus examination showed parafoveal reddish oval lesions, but they were mostly evident on infrared imaging.

Originally, AMN was considered a pathology of the inner retina layers, but more recent reports have demonstrated the location of the lesions in the outer retina, due to the widespread OCT usage [6,35]. Occasionally, a normalization of the retina layers some months after initial presentation was noted in the OCT images [35].

To date, it is still unknown the process causing the specific findings routinely noted in the outer retina, although there is now an increased understanding of both the chronology and imaging of the condition. Proposed mechanisms include an acute inflammatory process or vascular disease associated with systemic hypertension [17]. 

AMN lesions typically develop at the junction between the OPL and ONL and are often associated with disruption of the ellipsoid zone, similar to the OCT alterations of our patient. The most creditable hypothesis is that these changes result from local compromise of the DCP which provides retinal perfusion to the zone between the retinal and choroidal circulations, with a reperfusion in late stages [36]. This concept of DCP ischemia causing AMN is strongly supported by previously identified risk factors for AMN [2,16].

AMN results in a thinning of the affected retinal layers leading to residual scotoma, although in some cases complete visual recovery has been documented [37]. In our case, these retinal alterations were noted (left eye more than right one), but the patient showed an improvement in vision compared to the first visit. At the end of the follow-up, our patient complained about a scotoma in the left eye. Also, the infrared image showed attenuated lesions after two years. Visual field confirmed the paracentral scotoma in the left eye at the end of follow-up. No scotomas were noticed in the right eye visual field.

The OCT performed in our patient demonstrated an important role in defining chronological events in the different retinal layers. In our case, at the beginning, the OCT features were a focal highly reflective band in the OPL with thinning of the ONL and defects of the ellipsoid zone, which are consistent with previous studies [6,38,39].

Fawzi et al. [17], described a case of ONL thinning, attenuation of the ellipsoid zone and persistent absence of interdigitation zone in a patient after 14 months follow-up. Vance et al. [6], noted focal loss of the ellipsoid zone and ONL thinning with re-established features of the ellipsoid zone but a persistent ONL thinning. In our patient, the OCT showed hyperreflective spots in the OPL with a hyporeflective corresponding band of the ellipsoid zone, at the end of the follow-up.

## 4. Conclusions

This paper is a report of a bilateral case of AMN that demonstrated a different evolution in the two eyes of the same patient. Several mechanisms could be implicated in the pathology of AMN. However, we suspect that the retinas of both eyes were damaged through the same pathological mechanism but in a more severe grade in the left eye, such that the scotoma and IR/OCT alterations could not completely revert in that eye. OCT played a fundamental role in the AMN diagnosis and follow-up. It is still unclear whether a direct viral effect or immunologic phenomenon is responsible for changes of the retinal layers, therefore further studies are needed to investigate this rare pathology.

## Figures and Tables

**Figure 1 diagnostics-10-00259-f001:**
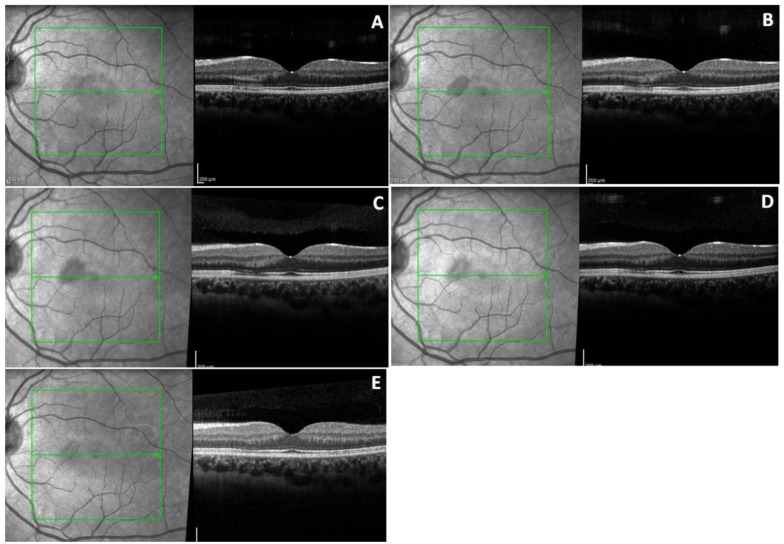
Optical coherence tomography (OCT) and infrared (IR) images of left eye. (**A**) At presentation IR shows hyporeflective parafoveal lesion. Corresponding OCT shows a hyperreflective band of outer plexiform layer (OPL) and outer nuclear layer (ONL), and alteration of inner segment/outer segment (IS/OS) and retinal pigment epithelium (RPE). (**B**) Imaging one week later. (**C**) 40 days later, the OCT features started to normalize. (**D**) One year later, the OCT shows a smaller band in the OPL. (**E**) 30 months later, the lesion on the IR image is reduced and there are hyperreflective spots in the OPL and hyporeflective band of the ellipsoid zone.

**Figure 2 diagnostics-10-00259-f002:**
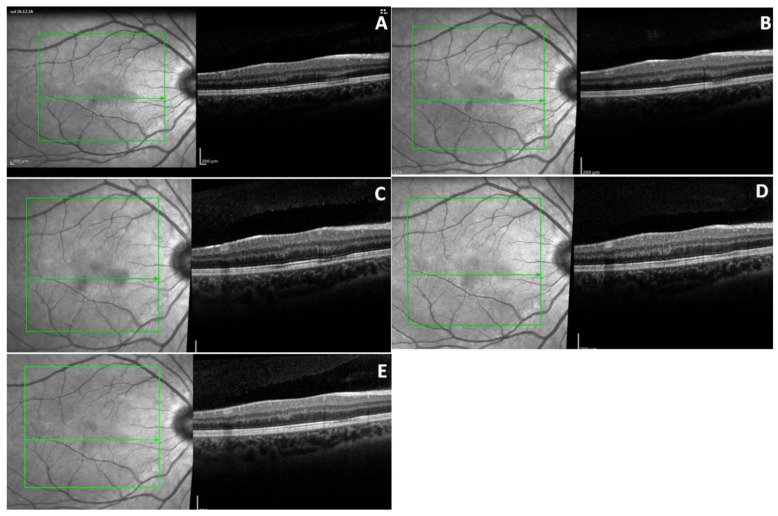
OCT and infrared images (IR) of right eye. (**A**) At presentation IR shows two hyporeflective parafoveal lesions. Corresponding OCT shows two hyperreflective band of outer plexiform layer (OPL) and outer nuclear layer (ONL), and alteration of inner segment/outer segment (IS/OS) and retinal pigment epithelium RPE in the region of the bigger lesion in IR (nasally). (**B**) Image one week later. (**C**) 40 days month later, the OCT features started to normalize. (**D**) One year later the OCT shows a smaller band in the OPL corresponding to the smaller lesion in IR (infero-temporally). (**E**) 30 months later the lesions on the IR image are strongly reduced and there are rare hyperreflective spots in the OPL and hyporeflective band of the ellipsoid zone.

**Figure 3 diagnostics-10-00259-f003:**
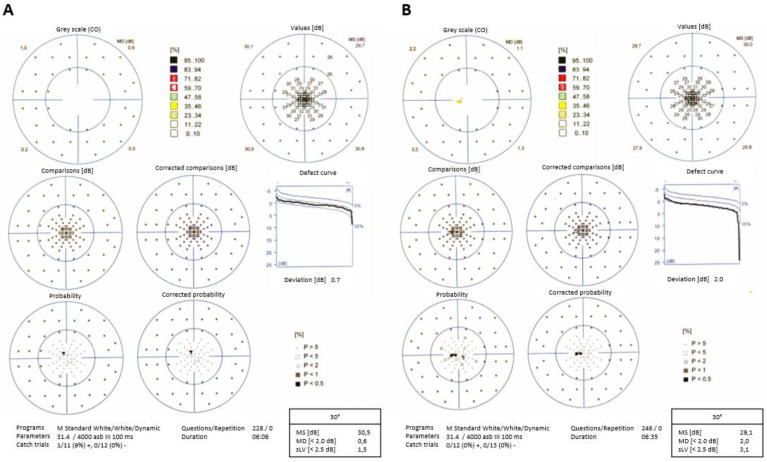
Visual field at presentation of right eye (**A**) and left eye (**B**) shows positive scotomas in the central region of both eyes.

**Figure 4 diagnostics-10-00259-f004:**
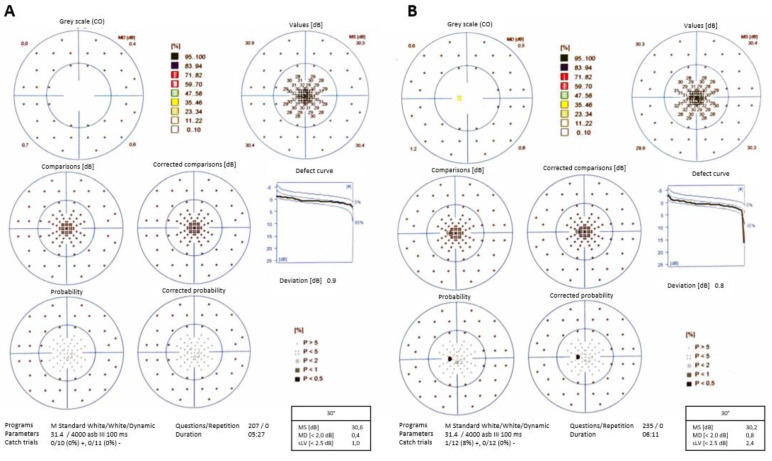
Visual field after 30 months shows disappearance of scotomas in right eye (**A**) and a residual paracentral scotoma in left eye (**B**).

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
