# Peer review of "Bilateral Acute Macular Neuroretinopathy in a Young Patient: Imaging and Visual Field during Two-Year-Follow-Up"

_diagnostics, 2020, doi:10.3390/diagnostics10050259_

Round 1
Reviewer 1 Report
This is a well documented case study of a bilateral acute macular neuroretinopathy (AMN) in a young female patient. The patient is particularly interesting as it does not have relation to any of the risk factors commonly associated with AMN (oral contraceptive, antecedent trauma, systemic shock, influenza vaccination etc). The only peculiar feature is the reported the flu-like symptoms the day prior the AMN presentation. Although association of AMN with flu-like illness or fever was previously speculated (Bhavsar et al., Surv Ophthalmol 2016, 61, 538-65.) the only thoroughly documented association is with dengue fever. The patient was subjected to number of clinical evaluations at baseline and during the follow-up: visual field, Optical Coherence Tomography (OCT), Fluorescein Angiography (FA) and Indocyanine Green Angiography (ICG). The methodologies are standard and the data collected with them are presented in clear fashion. What further made the case interesting is that a difference in the disease progression was observed in the two eyes (residual scotoma only in the left eye). As emphasized by the authors this suggest that several mechanisms are involved in AMN pathology. Furthermore the importance of OCT for the AMN diagnosis is well demonstrated.
I have minor recommendation related to the broader perspective of the discussion of the case study. At lines 150-152 it is mentioned that “One recent manuscript highlighted that mutations in more than 100 different genes may predispose to severe outcomes after influenza and this may be linked with the predisposition and pathogenesis of AMN [26].” Please provide a brief paragraph which summarizes the groups of genes and the main different mechanisms that might be involved. It will allow to put the case study in a broader context and will allow for the reader to critically think on the possible scenarios in which AMN may develop.
Author Response
.
Response to Reviewer 1 Comments
Point 1:
This is a well documented case study of a bilateral acute macular neuroretinopathy (AMN) in a young female patient. The patient is particularly interesting as it does not have relation to any of the risk factors commonly associated with AMN (oral contraceptive, antecedent trauma, systemic shock, influenza vaccination etc). The only peculiar feature is the reported the flu-like symptoms the day prior the AMN presentation. Although association of AMN with flu-like illness or fever was previously speculated (Bhavsar et al., Surv Ophthalmol 2016, 61, 538-65.) the only thoroughly documented association is with dengue fever. The patient was subjected to number of clinical evaluations at baseline and during the follow-up: visual field, Optical Coherence Tomography (OCT), Fluorescein Angiography (FA) and Indocyanine Green Angiography (ICG). The methodologies are standard and the data collected with them are presented in clear fashion. What further made the case interesting is that a difference in the disease progression was observed in the two eyes (residual scotoma only in the left eye). As emphasized by the authors this suggest that several mechanisms are involved in AMN pathology. Furthermore the importance of OCT for the AMN diagnosis is well demonstrated.
I have minor recommendation related to the broader perspective of the discussion of the case study. At lines 150-152 it is mentioned that “One recent manuscript highlighted that mutations in more than 100 different genes may predispose to severe outcomes after influenza and this may be linked with the predisposition and pathogenesis of AMN [26].” Please provide a brief paragraph which summarizes the groups of genes and the main different mechanisms that might be involved. It will allow to put the case study in a broader context and will allow for the reader to critically think on the possible scenarios in which AMN may develop.

Response 1:
We are grateful to the reviewer for the constructive comments on our manuscript. As suggested, we have implemented the manuscript adding the following paragraph in the Discussion section.
“According to Zhang et al., there are some candidate genes, whose proteins are implicated in the viral replication cycle and in host response stages.
More specifically, these genes could: 1) control the expression of oligosaccharide structures involved in binding of the virus to receptors on the host cell surface (es. DMBT1, MBL2, MUC1, SFTPA, SFTPB); 2) control cleavability of hemagglutinin by host proteases; 3) promote the viral replication into the host cell (phase 3 genes- NS1, PABPN1, EIF2C2, SiRNA); 4) lead to the activation of many pro-apoptotic genes (including those encoding Fas, TP53, JUN and CASP8); 5) up-regulate different INF-stimulated genes (such as CXCL10, MX1, PKR, OAS1, RNASEL, and PML); 6) and/or up-regulate the expression of molecules such as PTAFR (platelet activating factor receptor) used by bacteria as receptors so resulting in a bacterial over infection.
It seems that if these genes mutate, they could cause a severe increase of the inflammatory response, due to their role in the dysregulation of the host immunocompetence.”
Reviewer 2 Report
In this manuscript, Porta et al, describe a longitudinal study of a case of bilateral Acute Macular Neuroretinopathy (AMN) in a 21-year-old female, without known risk factors, that became symptomatic (fixed grey-dark spots in the vision field of both eyes) 1 day after experiencing flu-like symptoms. Fluorescein and ICG angiographies were all normal. Based on IR and OCT imaging, the authors describe the lesion in the left eye as more structurally important than in the right one. Scotoma, IR and OCT features disappeared at the end of the follow-up period (2 years) in the right eye but not in the left eye. I believe the manuscript will benefit if the following three issues are corrected:
- The conclusion: The authors conclude that this different disease progression in the two eyes of the same patient, suggests that several mechanisms are implicated in the pathology of AMN. However, a more likely scenario is that both left and right retinas were damaged through the same pathological mechanism but because the damage was more severe in the left eye, the scotoma and IR/OCT alterations could not completely revert in that eye. Accordingly, the authors should change or at least discuss more their conclusion.
- FIG 3 and FIG 4 should be simplified, the letter size should be increased and all remaining text translated to english.
- In the discussion, the sentences in lanes 134-136 need to be clarified
“Nevertheless, the only documented association is with the dengue fever [20, 21]. Other possible associations as citomegolovirus, influenza A2 and Chikungunga have shown weak evidence [22,23].”
Author Response
Response to Reviewer 2 Comments
In this manuscript, Porta et al, describe a longitudinal study of a case of bilateral Acute Macular Neuroretinopathy (AMN) in a 21-year-old female, without known risk factors, that became symptomatic (fixed grey-dark spots in the vision field of both eyes) 1 day after experiencing flu-like symptoms. Fluorescein and ICG angiographies were all normal. Based on IR and OCT imaging, the authors describe the lesion in the left eye as more structurally important than in the right one. Scotoma, IR and OCT features disappeared at the end of the follow-up period (2 years) in the right eye but not in the left eye. I believe the manuscript will benefit if the following three issues are corrected:
Point 1:
The conclusion: The authors conclude that this different disease progression in the two eyes of the same patient, suggests that several mechanisms are implicated in the pathology of AMN. However, a more likely scenario is that both left and right retinas were damaged through the same pathological mechanism but because the damage was more severe in the left eye, the scotoma and IR/OCT alterations could not completely revert in that eye. Accordingly, the authors should change or at least discuss more their conclusion.
Response 1:
Thank you for your constructive comment. We have implemented the conclusion according to your suggestion.
Point 2:
FIG 3 and FIG 4 should be simplified, the letter size should be increased and all remaining text translated to english.
Response 2:
Thanks for your comment. We have simplified, modified and translated the figures 3 and 4 according to your suggestions.
Point 3:
In the discussion, the sentences in lanes 134-136 need to be clarified.
“Nevertheless, the only documented association is with the dengue fever [20, 21]. Other possible associations as citomegolovirus, influenza A2 and Chikungunga have shown weak evidence [22,23].”
Response 3:
Thank you. The sentences have been re-writed to be clearer, as you suggested.
Reviewer 3 Report
This is an interesting case report of bilateral acute macular neuroretinopathy(AMN) in a young patient with 2 years followup.
The imaging provided are predominantly infrared, OCT images, and automatic visual fields. The 2 years followup are interesting.
The information provided are not new, as there have been number of publications on the etiology, and the image findings of the AMN. There is no new information provided in the case report.
The following are some edits which should be addressed:
ICGA is a better abbreviation for Indocyanine Green Angiography
On page 2, line 44, put the (SD-OCT) after spectral-domain optical…
On line 59, …or previous residence in tropical countries, suggest to add “or travel history”
For the legend of the Figure 1, 2, use “OCT “instead of “Oct”
On page 3, line 91, please explain the sentence “ An alteration of IS/OS interface and RPE was reported only in the zone of the bigger lesion.” Which bigger lesion?
There is a description of “bigger lesion” in the Figure 2 legend. I assume these are related.
On page 6, line 36, “citomegolovirus” should be “cytomegalovirus”
Line 141, deleted “would”
On page 4, line 09, and line 14, replace “reductions”, with “reduction”
On page 4, line 10, In addition, the “defected noted” should be “defect noted”, but it is unclear what defect was noted?
On page 6, line 67, replace “accredited” with “creditable”
Author Response
Response to Reviewer 3 Comments
Point 1:
This is an interesting case report of bilateral acute macular neuroretinopathy(AMN) in a young patient with 2 years followup.
The imaging provided are predominantly infrared, OCT images, and automatic visual fields. The 2 years followup are interesting.
The information provided are not new, as there have been number of publications on the etiology, and the image findings of the AMN. There is no new information provided in the case report.
Response 1:
Thank you for your time and constructive comments on our manuscript. However, as we had already reported in the discussion and conclusion sessions, our young patient had no risk factors and a bilateral AMN with a different evolution in the two eyes. Moreover, we performed the imaging and visual field in a long follow-up (two years).
Point 2:
The following are some edits which should be addressed:
1) ICGA is a better abbreviation for Indocyanine Green Angiography
2) On page 2, line 44, put the (SD-OCT) after spectral-domain optical…
3) On line 59, …or previous residence in tropical countries, suggest to add “or travel history”
4) For the legend of the Figure 1, 2, use “OCT “instead of “Oct”
5) On page 3, line 91, please explain the sentence “An alteration of IS/OS interface and RPE was reported only in the zone of the bigger lesion.” Which bigger lesion? There is a description of “bigger lesion” in the Figure 2 legend. I assume these are related.
6) On page 6, line 36, “citomegolovirus” should be “cytomegalovirus”
7)Line 141, deleted “would”
8)On page 4, line 09, and line 14, replace “reductions”, with “reduction”
9)On page 4, line 10, In addition, the “defected noted” should be “defect noted”, but it is unclear what defect was noted?
10)On page 6, line 67, replace “accredited” with “creditable”
Response 2:
Thanks for the comments that gave us the opportunity to improve the quality of our manuscript. We have addressed the edits as suggested by the reviewer.
On page 3, line 91 the term “bigger lesion” is referred to the IR image, and we add also the localization in the image. The legend in the figure 2 is related, and we provided to change it.
On page 4, line 10, we changed “defect noted” with the “hyporeflective area noted”, which is referred to the SLO image.
Round 2
Reviewer 3 Report
Please see highlighted comments in the pdf file of the manuscript.
The added paragraph provides explanation of the different way which gene interacts with the virus but lack explanation of how this is connected to AMN
